# The genome formula of a multipartite virus is regulated both at the individual segment and the segment group levels

Mélia Bonnamy[1,2], Andy Brousse[1,2], Elodie Pirolles[1], Yannis Michalakis[2]\*, Stéphane Blanc[1]\*

**1** PHIM, Univ Montpellier, IRD, CIRAD, INRAE, Institut Agro, Montpellier, France, **2** MIVEGEC, CNRS, IRD, Univ Montpellier, Montpellier, France

☯ These authors contributed equally to this work.
\* yannis.michalakis@ird.fr (YM); stephane.blanc@inrae.fr (SB)

**Data Availability Statement:** Files with results of qPCR reactions and scripts used for the statistical analyses are accessible at https://doi.org/10.5281/zenodo.10439590.

## Abstract

Differential accumulation of the distinct genome segments is a common feature of viruses with segmented genomes. The reproducible and specific pattern of genome segment accumulation within the host is referred to as the "genome formula". There is speculation and some experimental support for a functional role of the genome formula by modulating gene expression through copy number variations. However, the mechanisms of genome formula regulation have not yet been identified. In this study, we investigated whether the genome formula of the octopartite nanovirus faba bean necrotic stunt virus (FBNSV) is regulated by processes acting at the individual segment *vs.* viral population levels. We used a leaf infiltration system to show that the two most accumulated genome segments of the FBNSV possess a greater intrinsic accumulation capacity in *Vicia faba* tissues than the other segments. Nevertheless, processes acting at the individual segment level are insufficient to generate the genome formula, suggesting the involvement of additional mechanisms acting at the supra-segment level. Indeed, the absence of segments with important functions during systemic infection strongly modifies the relative frequency of the others, indicating that the genome formula is a property of the segment group. Together, these results demonstrate that the FBNSV genome formula is shaped by a complex process acting at both the individual segment and the segment group levels.

## Author summary

Segmented and multipartite viruses have their genomic information carried by several molecules, which allows for the unequal accumulation of the different genome segments in infected tissues. The reproducible and specific pattern of genome segment accumulation within the host is referred to as the "genome formula". The genome formula is host-dependent and believed to modulate gene expression through copy number variations upon host switches. The mechanisms leading to the genome formula remain unknown. Here, we determined that the genome formula of an octopartite single-stranded DNA

**Funding:** This work was funded by Agence Nationale pour la Recherche, ANR Reassort ANR-20-CE02-0016 (YM) and ANR Nanovirus ANR-18-CE92-0028 (SB), as well as by the University of Montpellier MUSE, project MULTIVIR (SB). EP, SB and MB acknowledge support from INRAE dpt. SPE; YM, AB and MB from CNRS, and MB from Univ Montpellier. The funders had no role in study design, data collection and analysis, decision to publish, or preparation of the manuscript.

**Competing interests:** The authors have declared that no competing interests exist.

nanovirus is shaped by processes acting both at the level of individual segments, some having higher accumulation rates, and at the level of the group of segments, the omission of some non-essential segments affecting the relative accumulation of others. Our study provides insights into the level at which genome formula regulation operates, giving a starting point for future studies aiming at elucidating the mechanisms governing genome formula in multipartite and segmented viruses.

## Introduction

Viruses can be monopartite, if their entire genetic information is carried by a single nucleic acid molecule; segmented, if their genomic information is carried by several molecules all encapsidated together, or multipartite, if their genomic information is carried by several genome segments which are packaged separately. Hence, in the first two types, the complete viral genome is packaged in a single virion while for multipartite viruses it is distributed over several virus particles.

A remarkable feature of segmented and multipartite viruses is that the fragmented nature of their genome enables the unequal accumulation of the different genome segments in infected tissues. This was first formalized for the multipartite single-stranded DNA (ssDNA) faba bean necrotic stunt virus (FBNSV), where some segments accumulate a lot while others are relatively rare [1]. Subsequently, the differential accumulation of genome segments in the host has been characterized for other multipartite DNA [2–6] and RNA [7] viruses and for segmented RNA viruses [8,9]. Because it was found in viruses with different genome nature and organization and infecting plants or animals, the differential accumulation of segments seems to be a common feature of viruses with fragmented genomes. The pattern of the frequency distribution of the distinct genome segments accumulation within the virus population was referred to as the genome formula [1]. Whenever investigated, the genome formula was shown to be host-specific [1,7,9,10] and the reproducible relative frequency distribution specific to a given host species was defined as the "set-point genome formula" [1].

Several authors speculated that the genome formula has a functional role in the regulation of viral gene expression by differentially modulating their respective copy numbers [1,11–13]. Similar processes, termed amplification-mediated gene expression tuning (AMGET) [14] or genomic accordion [15], were described for procaryotic and eukaryotic microorganisms, and large double stranded monopartite DNA viruses, where rapid adaptation of gene expression in new environments is primarily driven by the selection of copy-number variants [14–16]. Consistently, our group recently showed that the accumulation of FBNSV messenger RNAs is positively correlated to the accumulation of the corresponding genome segments, confirming that genome formula variations modulate gene expression [10].

Upon host-switching, the immediate and reversible host related changes of FBNSV genome formula are not induced by sequence modification [10]. At present, the mechanisms leading to the reproducible differential accumulation of the genome segments in a given host remain completely unknown. The genome formula could result from selection pressures acting on each segment independently, such as a difference in replication rates across segments [1]. Alternatively, or additionally, the establishment of the genome formula could be driven by selection acting at the level of the group of segments. It has been proposed that groups of segments with different frequency patterns could initiate distinct infection sites at early stages of host colonization, and those with the most optimal frequency pattern for the viral system would propagate faster and thereby be positively selected [1,11,12,17]. This is supported by

studies on RNA viruses showing that the very small number of genomes infecting a new cell may generate infection foci with different genetic compositions, increasing the potential for within-host between-foci selection. Adaptative genomes would then be selected, expand the viral infection and constitute the whole virus population [18,19]. In the case of viruses with fragmented genomes, each infection focus with a distinct genome formula could represent a copy number variant and the "set point genome formula" would result from selection of the better fit variant(s).

In this study, we used the FBNSV (family *Nanoviridae*, genus *Nanovirus*) to address the specific question of whether the genome formula is regulated by mechanisms acting on each individual segment or on a higher level involving more than one segment. FBNSV is a phloem-restricted phytovirus belonging to the genus of multipartite viruses with the highest number of genome segments known to date. Its genome is divided into eight circular ssDNA molecules (segments C, M, N, R, S, U1, U2, U4), each encapsidated separately [20]. All segments are assumed to replicate via rolling-circle replication (RCR) controlled by a viral protein of the Rep family, and they all possess a highly conserved replication origin [21–23]. Each segment is about 1 kb and encodes a single gene. C encodes the protein Clink which interacts with cell cycle regulators to enhance replication [24]. M encodes the movement protein (MP) [25]. N encodes a protein named "nuclear shuttle protein" (NSP) which was proved to be mandatory for aphid transmission [26,27]. R encodes the M-Rep protein which controls the RCR of all genome segments [28,29]. S encodes the capsid protein (CP) which encapsidates each segment individually [30,31]. U1, U2 and U4 encode proteins with unknown functions. Among the 10 FBNSV isolates collected in the field for which the entire genome has been sequenced and is available on NCBI, all eight segments were found in all cases. However, in experimental conditions it is possible to generate incomplete infections by omitting one or two of the "dispensable" segments at inoculation [27]. The impact of the absence of such segments on the infection phenotype depends on which segment is omitted. In *V. faba*, the absence of either C, N, or U4 does not impact the disease phenotype, although the absence of C reduces the infection rates by *Agrobacterium*-mediated inoculation. The absence of either U1 or U2 alters the timing and severity of symptoms development. M, R and S are essential.

Using a system of leaf infiltration with pairs of segments, each including R plus one of the seven others, we could estimate the local accumulation of individual segments out of the full genome context and in the absence of systemic movement. This showed that the intrinsic capacity to accumulate locally differs across segments, although segments do not accumulate at ratios that reproduce the systemic genome formula. Thus, processes at the segment level are likely acting, but are not sufficient to explain the genome formula found in systemically infected plants. Infiltrating leaves with all eight segments together did not locally reproduce the genome formula either, further suggesting that other processes occurring during systemic invasion are required. Finally, by deliberately omitting one or two dispensable segments in incomplete infections and analyzing the effect of their absence on the systemic accumulation of the others, we revealed that a group dynamic is also necessary to shape the set-point genome formula.

## Results

### The intrinsic capacity of the segments to accumulate in plant cells does not reproduce the genome formula

We first investigated whether the genome formula could result from differences in the intrinsic ability of each segment to accumulate in infected plant cells. To test this hypothesis, we quantified the accumulation of each FBNSV genome segment locally, *i.e.* in the absence of systemic

movement, and independently of the other segments except for segment R which encodes the protein M-Rep that is mandatory for replication [29]. Fully developed *V. faba* leaves were agro-infiltrated with copies of one of the seven segments C, M, N, S, U1, U2 and U4 along with segment R. The accumulation of each segment in infiltrated tissues was quantified by qPCR six days later. All segments were replicated when co-infiltrated with R, as verified with a replication-threshold estimate described in the Methods section, indicating that R is necessary and sufficient for the replication of all FBNSV segments.

We then compared the relative accumulation of each segment in infiltrated leaves to their respective frequency in systemic infections with the complete FBNSV genome. For this purpose, we calculated the ratio of the accumulation of each segment relative to the accumulation of R in both infiltrated and systemically infected leaves (Fig 1A). The normalization of the estimated concentration of each segment by that of R also permitted to account for possible variations of the infiltration efficiency, *i.e.* of the number of transfected cells and thus the total amount of replicated segments in distinct biological replicates. As expected, the accumulation ratios of the different segments in complete systemic infections differ greatly and nicely match with previous reports from our group (Fig 1, grey boxes; S1 Table) [1,10]. In infiltrated leaves, the ratios also differ across segments (Fig 1, blue boxes; S2 Table), showing that the genome segments accumulate unequally relatively to R in infiltrated tissues. For all segments, the ratios in infiltrated leaves are lower than those in systemic infections, indicating that R accumulates relatively more in the former condition. We believe this can be explained by a trivial bias of the

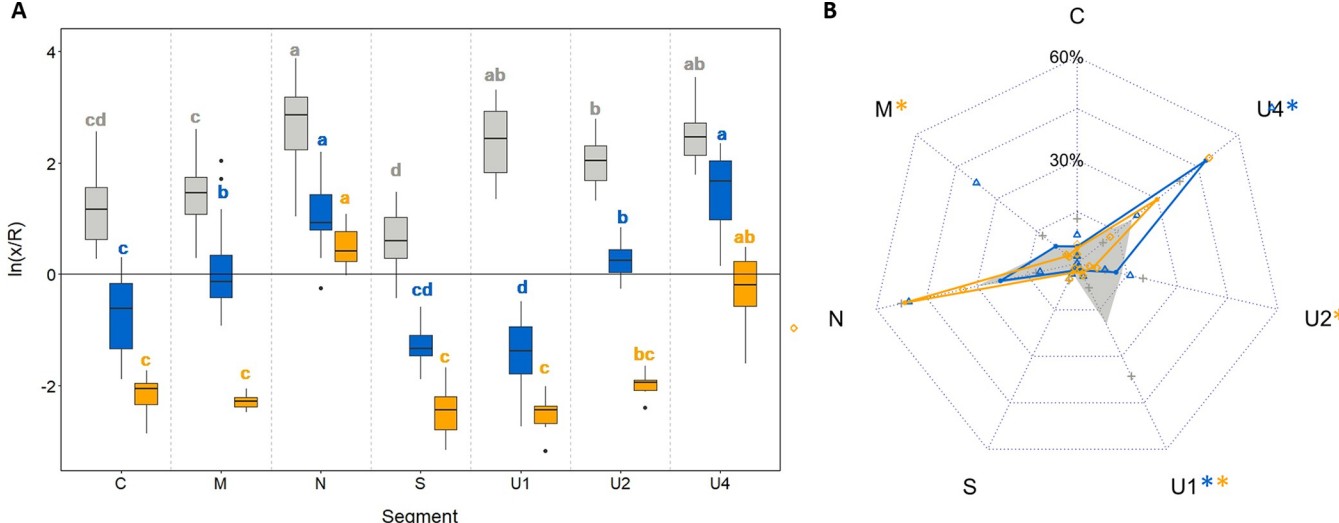

**Fig 1. FBNSV segment accumulation in systemically infected or infiltrated leaves of *V. faba*.** *V. faba* young plants were stem-agro-inoculated with the eight segments of the FBNSV. DNA was extracted from systemically infected newly formed leaves four weeks later (grey). Alternatively, *V. faba* leaves were agro-infiltrated with DNA-R and one of the other seven segments (blue) or with the eight segments (orange). DNA was extracted from infiltrated areas six days later. Virus DNA accumulation in infiltrated and systemically infected leaves was estimated by qPCR. Leaf infiltration data were obtained from two different experiments. (A) The relative accumulation ratio of each segment with respect to DNA-R was calculated. For each box, the horizontal central bar represents the median and the edges of the rectangle the first and third quartiles. The vertical outer bars delineate the minimum and maximum values of the distribution, excluding outliers. The dots represent outliers. For a given condition (infiltration in pairs, infiltration of all segments or systemic infection) the statistically significant differences between segments were assessed with Kruskal-Wallis tests (p≤0.05; Bonferroni correction) and are indicated by different letters in grey (systemic infections), blue or yellow (infiltrations). (B) Relative frequencies of the segments in samples shown in (A): systemic infection (grey), infiltrated with R and another segment (blue) and infiltrated with the eight segments (orange). Relative frequencies in systemic infections match well with previous reports [1,10]. Standard deviations are represented by grey crosses (systemic infections), blue triangles (infiltration with R) or orange squares (infiltrations with the eight segments). Asterisks associated to segment names indicate when the differences in frequencies between systemic infections and infiltrations in pairs (blue) or infiltration of all segments (orange) are statistically significant. Differences were assessed with the Scheirer Ray Hare (p≤0.05) and post-hoc Dunn tests (Bonferroni correction).

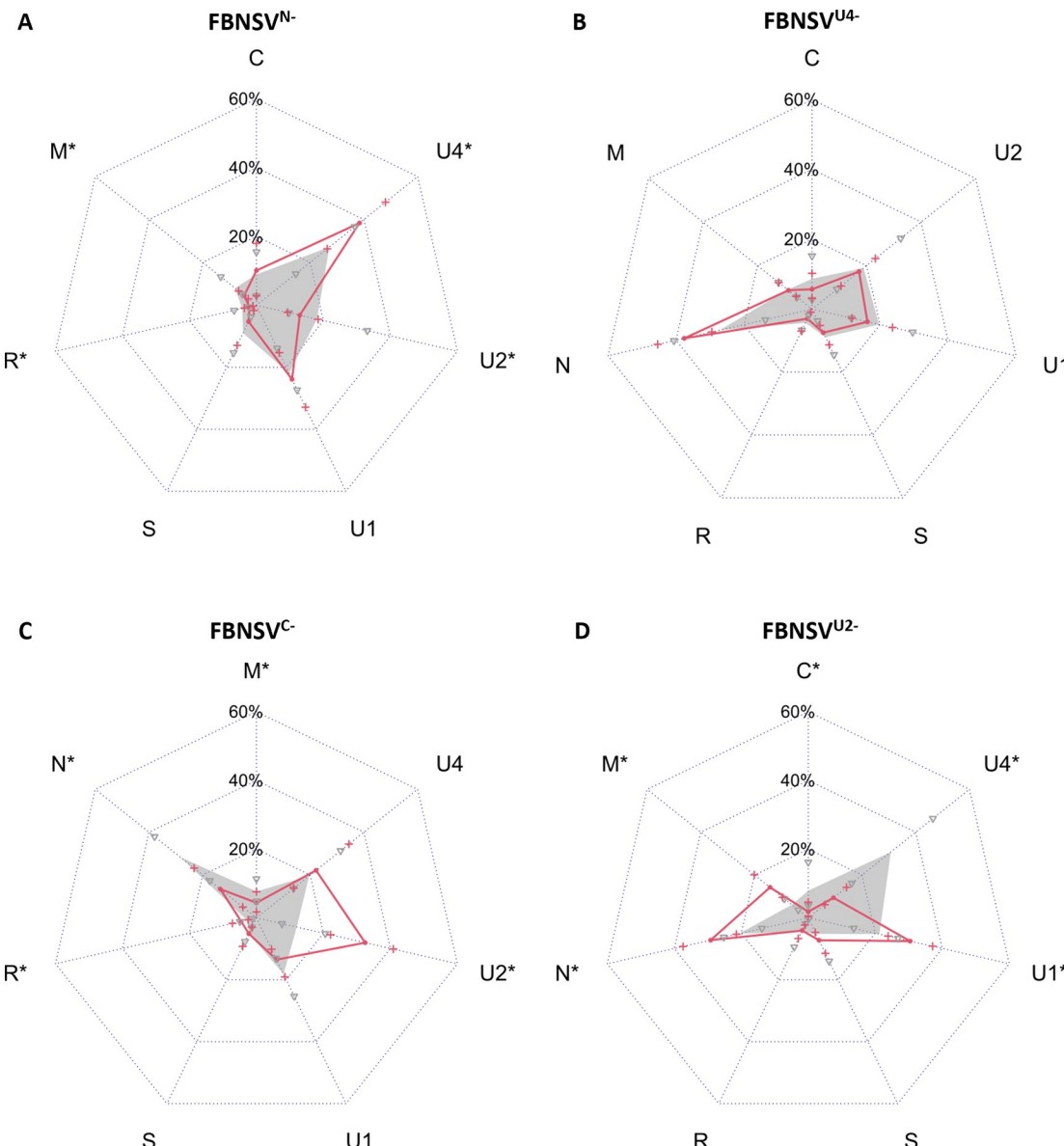

**Fig 2. Comparison of the FBNSV genome formula in complete and incomplete infections in *V. faba*.** FBNSV genome formula in complete infections (FBNSV^complete) is shown in grey whereas that in incomplete infections is shown in red for: (A) FBNSV^N-, (B) FBNSV^U4-, (C) FBNSV^C- or (D) FBNSV^U2-. Genome segments accumulation in symptomatic plants has been estimated by qPCR and the relative frequency of the segments was determined. To allow meaningful comparisons, for each radar plot the relative frequency of each segment in complete infections has been calculated without considering the accumulation of the segment which is omitted in incomplete infections. Standard deviations are represented by grey triangles (complete infections) or red crosses (incomplete infections). Asterisks associated to segment names indicate when the differences in frequencies between complete and incomplete infections are statistically significant (Scheirer Ray Hare (p≤0.05) and post-hoc Dunn tests, Bonferroni correction).

infiltration method, as further commented in the Discussion section. More importantly, the patterns of the segment ratios are different in infiltrated and in systemically infected leaves. In both infiltrated and systemically infected tissues, N and U4 are the most accumulated segments while S is the least. However, patterns differ for the other segments, the most striking being U1 whose relative accumulation switches from very low in infiltrated leaves to very high in systemic infection.

To better visualize and compare the segment accumulation patterns between infiltrated and systemically infected leaves, we reconstructed a genome formula from the leaf infiltration results: for each segment we calculated the median of the ratios obtained for each sample divided by the sum of the medians of the ratios obtained for each segment (Fig 1B and S3 Table). The accumulation pattern in agro-infiltrated leaves (blue) is different from that of systemically infected leaves (grey), confirming that individual segment accumulation in infiltrated cells does not reproduce the genome formula that develops upon systemic infection of the host plant by the complete FBNSV genome. To determine whether this difference is due to our experimental design where the local accumulation of each segment in infiltrated tissues is considered independently of the other segments, we co-infiltrated the eight FBNSV genome segments together in the same leaves. Unfortunately, for technical reasons related to the agroinfiltration of a mixture of eight distinct bacterial cultures (see Materials and Methods), we obtained only a small number of leaves (six out of twenty infiltrated leaves) with all eight segments amplified by qPCR. In these few leaves, the accumulation pattern proved different from the systemically infected plants, but not from the leaves infiltrated by single segments and R (Fig 1; orange; S3 and S4 Tables). Again, N and U4 are the most accumulated segments, but the six other segments show very low accumulation relative to R.

In summary, the leaf infiltration experiments show that N and U4, which are the most frequent segments in systemic infections, also accumulate at higher levels locally and individually. Thus, the intrinsic capacity of these segments to accumulate appears to contribute to the genome formula. However, segment accumulation in infiltrated leaves does not entirely reproduce the genome formula, indicating the involvement of a regulatory mechanism occurring at another scale. We thus investigated the segment accumulation pattern at the whole plant scale in the next section.

## The genome formula is shaped by mechanisms acting on the segment group during systemic infections

Since the basic capacity of the segments to accumulate locally (individually or together) is not sufficient to explain the genome formula, we investigated their respective accumulation at higher scales, by considering the group of segments and systemic infections. We exploited the possibility to produce incomplete infections [27] by inoculating *V. faba* plantlets with only seven, and in one case six segments. Total DNA was then extracted from systemically infected leaves and the genome formula was determined by calculating the frequency of each segment within the total virus population [1]. We chose to test this in two *a priori* different modalities: i) absence of a segment that does not affect infectivity by agro-inoculation and/or the timing and intensity of symptoms, and ii) absence of a segment that affects at least one of the previously cited traits [27].

Consistent with an earlier report [27], plants infected with FBNSV by all segments except segment N (FBNSV$^{N-}$) or U4 (FBNSV$^{U4-}$) exhibited similar time of symptom appearance and similar symptom severity as those infected with all segments (FBNSV$^{complete}$) (S5 Table). The genome formula of FBNSV$^{N-}$ differed statistically significantly from that of FBNSV$^{complete}$ for segments M, R, U2 and U4 even though the mean differences were relatively small (Fig 2A and S6 Table). The genome formula in plants infected with FBNSV$^{U4-}$ was not statistically significantly different from that in FBNSV$^{complete}$ (Fig 2B and S6 Table). For both conditions, the segment accumulation pattern is qualitatively similar to that of FBNSV$^{complete}$. Hence, the absence of segments N or U4 does not have a qualitative impact on the accumulation of the other segments in our experimental conditions.

Again, as earlier reported [27], the omission of segment C (FBNSV$^{C-}$) and even more of U2 (FBNSV$^{U2-}$) led to a decrease of the infection rate in agro-inoculated *V. faba* plants (S5 Table).

FBNSV$^{C-}$ showed no differences in the timing of appearance and in the severity of symptoms compared to FBNSV$^{complete}$, while symptoms were delayed and attenuated for plants infected with FBNSV$^{U2-}$ (S1 Fig and S5 Table). The genome formulas of both FBNSV$^{C-}$ and FBNSV$^{U2-}$ were modified compared to FBNSV$^{complete}$. In comparison to plants infected with FBNSV$^{complete}$ all segments but S, U1 and U4 were statistically significantly impacted in plants infected with FBNSV$^{C-}$ (Fig 2C and S6 Table), and all segments but R and S were statistically significantly affected by the absence of U2 (Fig 2D and S6 Table). Hence, the absence of C or U2 differentially modifies the accumulation of the other segments quantitatively and qualitatively, and their relative accumulation pattern depends on the identity of the missing segment.

We also infected plants omitting both C and U4 segments to see the effect on genome formula of the concomitant absence of a segment which does not modify the genome formula (U4) and a segment which does (C). Plants infected with FBNSV$^{C-, U4-}$ showed similar time of symptom appearance and symptom severity compared to FBNSV$^{complete}$. Interestingly, the genome formula in plants infected with FBNSV$^{C-, U4-}$ was not statistically significantly different from that in plants infected with FBNSV$^{C-}$ and the pattern of accumulation was qualitatively conserved (S2B Fig and S6 Table), confirming a low impact of U4 on the relative accumulation of other segments.

Overall, our results demonstrate that the absence of a segment differentially affects the frequency of the others. The remaining group of segments apparently adopts a novel set-point genome formula upon systemic infection. Interestingly, this phenomenon does not occur when the missing segment is N or U4, which have no impact on the disease development, but are highly accumulated during *V. faba* infection.

## Discussion

We first determined that segment R is sufficient for the replication of any FBNSV genome segment in agro-infiltrated *V. faba* leaves. This was expected since Timchenko and collaborators demonstrated that pairwise infiltrations of each segment of the nanovirus faba bean necrotic yellows virus (FBNYV) with R is sufficient to induce their replication in *Nicotiana benthamiana* leaf discs [28,29]. We thus verified here that for FBNSV as well R is the only segment mandatory for the replication of all genome segments *in planta*.

More surprising was the observation that, relative to the co-infiltrated segment R, FBNSV segments do not accumulate equally. These differences between individual segments could result from segment-specific replication rates. Considering that all segments have approximatively the same size, different replication rates should be due to distinct regulatory processes. Two elements are thought to be of primary importance for nanovirus replication: the iterons which determine the recognition of viral DNA by the Rep protein, and a sequence that forms a stem-loop structure and constitutes the replication origin. The FBNSV iteron sequences are identical for the eight segments. The stem sequence is identical for segments C/U1 and for segments M/N/R/S/U2/U4, but slightly differs between these two groups to which it may confer a different stem-loop stability. However, these sequence variations are not consistent with the results obtained from the leaf infiltration experiments because segments of the same stem sequence group (for example S and U4) differ greatly in their accumulation levels. Differential replication rates could be caused by other yet unidentified regulatory sequences or by differences in the synthesis of the viral complementary strand which serves as the template for replication [23]. Alternatively, accumulation differences between segments could result from differential degradation of the DNA segments due to the nature of their sequence and secondary structures. Another possibility could be that interactions between proteins resulting from the expression of the infiltrated segments could affect segment ratios. Among other things,

protein-protein interactions could lead to a different accumulation of segment R depending on the segment with which it is infiltrated. When comparing the accumulation of segment R infiltrated with each of the other segments or alone, we found that overall segment R accumulated at equivalent amounts. The only statistically significant difference was found between infiltrations with segment U1 (more accumulated) *vs*. segment U2 (less accumulated) (S3 Fig and S8 Table). Hence, segment R appears to accumulate in an equivalent way whatever the segment with which it is infiltrated, and its putative differential accumulation when paired with different segments cannot explain the higher accumulation on N and U4 for example.

When taken individually, though the distinct segments accumulate at different concentrations, their normalized relative accumulation pattern is not consistent with that observed when they systemically infect hosts together. First, R is much more accumulated than other segments in infiltrated compared to systemically infected leaves (Fig 1A). Two characteristics of plant systemic infections by the FBNSV must be considered to tentatively interpret this observation: FBNSV is naturally restricted to the companion cells [32], and its genome segments rarely co-localize in individual cells which implies a cross complementation by intercellular movement of gene expression products [33]. In leaf infiltration experiments, the infiltrated cell types are not identified but are most likely mesophyll cells. Obviously, FBNSV can replicate in this type of cells, but it is likely that there are no intercellular movements of viral products (DNA, RNA or proteins) as these are naturally moving in and out solely of sieve elements and companion cells. A series of three studies in the genera *Babuvirus* and *Nanovirus* on the tissue specificity of the promotor of segment R [34–36] indicated some degree of specificity for the vascular cells. In our case, however, because the simultaneous co-infiltration of the eight FBNSV segments in fully developed leaves never led to systemic infections, it seems unlikely that the infiltrated segments often reached the vascular bundles. The consequence of this artificial leaf-infiltration system is thus that complementation between segments may be possible only if they penetrate the same cell. Any segment penetrating a cell in the absence of R would not be able to replicate, while R alone can. Thus, the relative accumulation of R measured in infiltrated leaves may well be overestimated compared to that of the other segments and most likely does not reflect an intrinsically higher accumulation. Despite this obvious possibility, the other segments inoculated in pairs with R should *a priori* all have the same chance to penetrate a cell in which R is present and so their accumulation should be comparable. When comparing them, their accumulation pattern does not match that observed upon systemic infection (Fig 1). This could have several explanations: (i) the infected cell type, (ii) the absence of interaction between the different segments beyond that with R in leaves infiltrated only by R and another segment, or (iii) the local scale precluding successive replication cycles when colonizing newly formed tissues.

Relative to the first potential explanation, in infiltrated leaves the FBNSV segments are mainly replicated in mesophyll cells, and perhaps in a few epidermic cells, while during systemic infections the virus is phloem-limited. It is possible that the cell type impacts the relative accumulation of segments. However, as stated above and due to near identical regulatory sequences, it seems unlikely that the cell type can impact differentially the accumulation of each segment. To our knowledge, this has not been investigated and we do not see an immediate feasible approach to test for it.

With respect to the second potential explanation, by co-infiltrating each segment solely with R we were interested in the ability of each segment to accumulate independently of the others. Nevertheless, when together, interactions between segments could impact their accumulation. For example, there could be competition between segments for replication, or the expression product of a given segment could affect the accumulation of another (including R). Viral DNA-protein or protein-protein interactions have been identified in some members of

the family *Nanoviridae* [25,37–39], but their relevance for the replication of different genome segments has not been investigated. To study whether interactions between segments (or their expression products) could affect their accumulation locally, we co-infiltrated the eight FBNSV genome segments together in the same leaf. The pattern of segment accumulation when the eight segments are infiltrated together is not statistically significantly different from that when infiltrated solely with R, and differs from that in systemically infected leaves (Fig 1). Segments N and U4 appear highly accumulated in all three experimental conditions, which may suggest that they are more competent for replication. However, the interpretation of the results obtained with co-infiltration of the eight segments should be very cautious. Indeed, it is likely that each cell does not necessarily internalize all segments together, and the probable lack of movement of viral material across distinct cells makes the interactions between the segments (or their expression products) uncontrollable. Therefore, apart from the high accumulation of U4 and N, we cannot draw further sound conclusions on the impact of inter-segment interactions on their local accumulation.

As for the third potential explanation, the leaf infiltration system allows to study segment accumulation individually, and probably at a cellular level as discussed above. However, the genome formula could result from mechanisms operating at a supra-cellular level. Our group [40] earlier estimated that the effective population sizes of two FBNSV segments (S and N) during transmission by the vector and plant colonization is relatively low. The low number of genomic segments propagating within hosts and initiating new infection foci could generate random variation of the frequency of genome segments across foci. This phenomenon designated genome formula drift [11,13] could enhance selection at the between foci level [18,19] on the genome formula. During successive cycles of infection of new cells or tissues, the infection foci that would multiply the most would be those for which the genome formula is closer to the optimal, rapidly driving the virus population towards the set-point genome formula [1,11,13]. Would this hypothesis be valid, then the set-point genome formula could be different from the "local genome formula", where no successive selection cycles occur, and where the formula is determined solely by the capacity of each segment to accumulate during one replication round. In addition, the viral system as a whole would adapt to environmental changes by adopting a new optimal formula.

To further investigate segment accumulation at the systemic level, we determined the set-point genome formulas in incomplete infections. The absence of a segment that has no impact on the infection (N or U4) does not qualitatively affect the relative frequencies of the remaining segments. Conversely, when the absence of a segment negatively affects FBNSV infection of *V. faba* (C or U2), this absence qualitatively impacts the accumulation of other segments that must collectively adjust to a novel set-point formula (Fig 2). These conclusions support the hypothesis of a selection at the level of the segment group [1,11,17].

The absence of N or U4 does not qualitatively impact the accumulation of the other segments, and interestingly, they are also distinguished from other segments by the fact that they have a greater intrinsic capacity to accumulate in *V. faba* infiltrated cells. N is required for aphid vector transmission [26,27], but its potential involvement in other steps of the viral cycle is unclear. It has been proposed that the NSP of the babuvirus BBTV may assist in the transport of viral proteins or DNA out of the nucleus [37], but its absence does not appear to impact FBNSV infection *in planta*, and its function during transmission may be effective solely during the crossing of aphid vector gut or salivary gland cells [26,41]. The function of the U4 protein is entirely unknown, and it has been reported repeatedly to have no within-host function, at least in experimental conditions. If these two segments are totally dispensable *in planta* and have even no impact on the genome formula, and therefore on the other segments, they could potentially get lost upon successive bottlenecks during progression into uninfected and newly

formed tissues. Because in all the experiments reported in this study they both consistently appear to be highly accumulated, we could speculate that they have unknown regulatory sequences which make them more replication/accumulation competent than the others. This could have been selected to avoid their loss since at least N is mandatory to complete the natural life cycle.

In conclusion, the genome formula of FBNSV seems to be determined for one part by regulation at the segment level, especially for the very frequent N and U4, and for another part by processes acting at the supra-segment level. Our results thus support the hypothesis that the unit of selection for the genome formula is at a supra-segment level.

## Materials and methods

### Plants, growth conditions, and FBNSV infectious clone

Faba beans (*Vicia faba*) var. Seville (Vilmorin) were grown in a growth chamber with 13h/11h day/night photoperiod, 25/18°C day/night temperature.

The FBNSV infectious isolate JKI-2000 used in this study was cloned by [20]. The infectious clone is constructed as eight binary plasmids (derived from pBIN-19), each containing a tandem repeat of one of the segments. Each of these plasmids was used to transform one colony of *Agrobacterium radiobacter* (formerly called *Agrobacterium tumefaciens*) strain COR308, yielding eight bacterial clones each capable of transferring one of the eight segments (C, M, N, R, S, U1, U2, U4) to a host plant upon agro-infiltration.

### Leaf agro-infiltrations

The eight *A. radiobacter* clones were separately grown in NZY+ medium (0.1% NZ amine, 0.5% yeast extract, 0.5% NaCl at pH7.5) completed with 12.5 mM MgCl2, 12.5 mM MgSO4 and 0.4% glucose, 10 mM MES pH 5.5, 50 μM acetosyringone, 50 mg.ml$^{-1}$ kanamycin, 25 mg. ml$^{-1}$ gentamycin and 5 mg.ml$^{-1}$ tetracycline, at 28°C with stirring at 150 rpm for 16 h. Bacteria were then pelleted by centrifugation and resuspended in 10 mM MgCl$_2$ solution. Different mixes of agrobacteria solutions were prepared depending on the experimental design: each segment alone (optical density (OD) = 0.4), mixes of pairs of segments containing R (OD = 0.4) and one of segments C, M, N, S, U1, U2, or U4 (OD = 0.4), or mixes containing R (OD = 0.4) and all the seven other segments (OD = 0.06 for each). Each mix was infiltrated on the underside of 14 days-old *V. faba* plantlets fully expanded leaves.

### Systemic infections

Systemic infections were obtained by agro-inoculating *V. faba* plants based on the protocol described earlier [1]. *A. radiobacter* solutions were prepared as described for leaf agro-infiltration experiments and were inoculated by picking and injecting into the stem of nine days old seedlings. Complete infections (FBNSV$^{complete}$) were made by inoculating a mix of the eight *A. radiobacter* clones, re-constituting the full FBNSV infectious clone. For incomplete infections, only seven (FBNSV$^{C-}$, FBNSV$^{N-}$, FBNSV$^{U2-}$, FBNSV$^{U4-}$) *Agrobacterium* clones were mixed and inoculated.

For plants inoculated by FBNSV$^{U2-}$ only one infected plant was obtained by *Agrobacterium*-mediated inoculation. This plant was used as a source to inoculate other plants by aphids (*Acyrthosiphon pisum*, clone 210). After a three days acquisition access period (AAP) on the infected plant (four weeks post agro-inoculation), viruliferous aphids were transferred on eleven days old *V. faba* plants (ten aphids per plant). After an inoculation access period (IAP) of two days, aphids were collected, mixed, and used to inoculate a second batch of thirteen

days old plants for a two-day IAP (eight or nine aphids per plant). After this second inoculation, the two batches of plants were treated with Pirimor G (1 g.L$^{-1}$ in water) to kill aphids.

Plants infected with FBNSV$^{C-, U4-}$ were obtained by inoculating eight days old *V. faba* plantlets with one aphid (*A. pisum*) that fed first on FBNSV$^{C-}$ for three days, then on FBNSV$^{U4-}$ for three more days or *vice versa*. After three days of inoculation, plants were treated with Pirimor. Plants in which neither C nor U4 were transmitted were used for this study.

## DNA extraction and qPCR conditions

For infiltrated leaves, total DNA was extracted from infiltrated tissues six days after infiltrations following the protocol described by [42]. For systemically infected plants, total DNA was extracted from symptomatic apical leaves three (FBNSV$^{C-}$ and FBNSV$^{C-, U4-}$) or four (FBNSV$^{N-}$, FBNSV$^{U2-}$ and FBNSV$^{U4-}$) weeks after inoculation following the same protocol. The concentration of total extracted DNA was estimated using a spectrophotometer Nano-Drop 2000 (Thermo Scientific, Waltham, MA, USA) and that of each genome segment was then determined by qPCR using the LightCycler FastStart DNA Master Plus SYBR Green I kit (Roche, Indianapolis, Ind, USA). Following the manufacturer's instructions, 5 μL of the 2X qPCR Mastermix were mixed with segment-specific primers (S7 Table) at 0.3 μM (segments C, M and S) or 0.5 μM (segments N, R, U1, U2, U4) final, 2 μL of 10-fold diluted DNA extracts and complemented with water to obtain a final reaction volume of 10 μL. qPCR reactions were carried out in a LightCycler 480 thermocycler (Roche) with 40 cycles of 95°C for 10 s, 60°C for 10 s and 72°C for 10 s. Two technical replicates were done for each sample. Post-PCR data analyses were carried out with the LinRegPCR software [43].

## Data analysis

**Leaf infiltration experiments.** For each sample, the DNA concentration measured by qPCR was normalized by the concentration of total DNA extracted from the leaves. Thresholds were then determined to remove samples where FBNSV segments have not been replicated in presence of segment R. Samples with concentrations below the threshold were excluded from the analysis as we considered that the segment was not replicated.

To estimate the thresholds, we quantified virus DNA segments from ten leaves infiltrated with either of the seven segments (C, N, S, U1, U2, U4) alone, without R. We could thus determine the qPCR basal level measured in the absence of replication (no segment R). We then calculated the upper tolerance threshold for each FBNSV segment such that 95% of potential future samples in which the segment is absent would have lower estimated DNA concentration with a 95% probability, based on the method described by [44,45]. The thresholds were 1.990203e-06 for segment C, 5.402976e-06 for M, 3.799221e-06 for N, 3.068326e-06 for S, 1.709007e-06 for U1, 3.235822e-06 for U2 and 3.716083e-06 for U4.

Because in some cases the analysis of variance assumption of homoscedasticity and/or normal distribution of residuals were not respected even after transforming the data we opted for simplicity to perform all the analyses using non-parametric tests on untransformed variables. We note, however, that the results of these tests match closely those of equivalent parametric tests even when the analysis of variance assumptions do not hold (results not shown).

To compare the accumulation of each DNA segment in agro-infiltrated leaves (in pair with R or with all segments) to their respective accumulation in systemic infections, the concentration of each individual segment was divided by that of R in the same sample. Statistical analyses were then performed through Kruskal-Wallis tests (p≤0.05; Bonferroni correction) using RStudio (package "agricolae"). For segment accumulation in leaves infiltrated with pairs of

segments, data were obtained from two independent experimental replicates. To allow for the statistical comparison with systemic infections and because the sample size of one experimental replicate for every segment was <10, and for some as small as three or four, we pooled the data of the two experimental replicates. To assess whether this was reasonable, the data obtained for the two experimental replicates were compared through Scheirer Ray Hare tests (p≤0.05) using RStudio (package "rcompanion"). The interaction segment * replicate was not statistically different, indicating that the accumulation of the segments in the two experimental replicates was similar (S4 Fig and S9 Table). Because of this, and because unduly pooling the replicates could only increase the residual variance, and thus make it more difficult to detect differences due to other treatments, the results of the two replicates were pooled for comparison with the other modalities.

Genome formulas were reconstructed from the results of agro-infiltrations and were compared to the genome formula in systemic infections. For each sample, the accumulation of each segment normalized by that of R was divided by the sum of the medians across all samples of the accumulation of each segment normalized by R. Statistical analyses were performed through Scheirer Ray Hare tests (p≤0.05) and post-hoc Dunn tests using RStudio (packages "rcompanion" and "FSA").

**Incomplete infections experiment.**   Plants with one or more missing segments other than those voluntarily omitted were excluded from the analysis. The frequency of each segment was expressed relatively to the total virus DNA for incomplete infections experiments. To compare the frequencies in incomplete and complete infections, the relative frequency of the segments in complete infections was calculated without considering the accumulation of the segment which is removed in incomplete infections. Statistical analyses were performed through Scheirer Ray Hare tests (p≤0.05) and post-hoc Dunn tests using RStudio (packages "rcompanion" and "FSA"). Results were obtained from one (FBNSV$^{U2-}$, FBNSV$^{C-, U4-}$), two (FBNSV$^{N-}$, FBNSV$^{U4-}$) or three (FBNSV$^{C-}$) independent experimental replicates. Data and scripts are available at https://doi.org/10.5281/zenodo.10439590

## Supporting information

**S1 Fig.**  Symptoms of *V. faba* infected with FBNSV$^{complete}$ (A and C) or FBNSV$^{U2-}$ (B and D) four weeks after agro-inoculation.
(TIF)

**S2 Fig. Comparison of the FBNSV genome formula in complete and incomplete infections without C and/or U4.** FBNSV genome formula in incomplete infections FBNSV$^{C-, U4-}$ (blue) compared to FBNSV$^{complete}$ (A; grey) or FBNSV$^{C-}$ (B; red). Genome segments accumulation in symptomatic *V. faba* plants was estimated by qPCR and the relative frequency of the segments was determined. To allow meaningful comparisons, the relative frequency of each segment was calculated without considering the accumulation of segments C and U4 in FBNSV$^{complete}$ and without considering U4 in FBNSV$^{C-}$. Standard deviations are represented by grey triangles (complete infections) or red crosses (incomplete infections). Asterisks associated to segment names indicate when the differences in frequencies between complete and incomplete infections are statistically significant (Scheirer Ray Hare (p≤0.05) and post-hoc Dunn tests, Bonferroni correction).
(TIF)

**S3 Fig. Comparison of DNA-R accumulation depending on the segment with which it was infiltrated.** The accumulation of DNA-R when infiltrated with each of the seven other segments (C+R, M+R, N+R, S+R, U1+R, U2+R or U4+R) or alone with the same (R) or doubled

(R+R) OD of infiltrated bacteria was determined by qPCR. For each box, the horizontal central bar represents the median and the edges of the rectangle the first and third quartiles. The vertical outer bars delineate the minimum and maximum values of the distribution, excluding outliers. The dots represent outliers. Letters above the boxes indicate segments with which DNA-R accumulation is statistically significant (Kruskal-Wallis tests and Bonferroni correction for multiple tests; S8 Table).
(TIF)

**S4 Fig. Comparison of segment accumulation when infiltrated in pairs with DNA-R across experimental replicates.** The relative accumulation ratio of each segment with respect to DNA-R obtained for the two experimental replicates are represented in dark and light blue respectively. For each box, the horizontal central bar represents the median and the edges of the rectangle the first and third quartiles. The vertical outer bars delineate the minimum and maximum values of the distribution, excluding outliers. The dots represent outliers. No statistically significant differences were found between the two experimental replicates (Scheirer Ray Hare test, $p \leq 0.05$; S9 Table).
(TIF)

**S1 Table. Statistical analysis of the comparison of segment accumulation relative to R in systemically infected leaves.** Comparisons of the accumulation of each segment relative to R to that of the others in systemically infected leaves were performed through Kruskal-Wallis tests using RStudio (package "agricolae"). The p-value indicating a statistically significant difference after Bonferroni correction ($p \leq 0.05$) is in red.
(DOCX)

**S2 Table. Statistical analysis of the comparison of segment accumulation relative to R in leaves infiltrated with each segment in pair with R.** Comparisons of the accumulation of each segment relative to R to that of the others in leaves infiltrated with pairs of segments were performed through Kruskal-Wallis tests using RStudio (package "agricolae"). The p-value indicating a statistically significant difference after Bonferroni correction ($p \leq 0.05$) is in red.
(DOCX)

**S3 Table. Statistical analysis of the comparison of segment accumulation according to the condition.** For comparisons of the accumulation of each segment relative to R in each modality (systemic infections, leaf infiltration with pairs of segments or infiltrations with the eight segments), we first provide the output of a full model, ratio = modality * segment. This analysis was performed through Scheirer Ray Hare tests using RStudio (package "rcompanion"). After the full model tests we provide the output of per segment comparisons across pairs of modalities to identify segments whose relative frequency statistically significantly differed between modalities. The per segment differences were assessed through Dunn tests using RStudio (package "FSA"). The p-values indicating statistically significant differences after Bonferroni correction ($p \leq 0.05$) are in red.
(DOCX)

**S4 Table. Statistical analysis of the comparison of segment accumulation relative to R in leaves infiltrated with the eight FBNSV segments.** Comparisons of the accumulation of each segment relative to R to that of the others in leaves infiltrated with the eight segments were performed through Kruskal-Wallis tests using RStudio (package "agricolae"). The p-value indicating a statistically significant difference after Bonferroni correction ($p \leq 0.05$) is in red.
(DOCX)

**S5 Table. Infection rates and phenotypes of FBNSV complete and incomplete infections in *V. faba*.** Infection rates and symptom severity were determined three (FBNSV$^{complete}$, FBNSV$^{N-}$, FBNSV$^{U4-}$, FBNSV$^{C-}$ and FBNSV$^{C-,U4-}$) or four (FBNSV$^{U2-}$) weeks after inoculation. The presence of the segments was controlled by qPCR.
(DOCX)

**S6 Table. Statistical analysis of the comparison of the segment relative frequency between complete and incomplete infections.** For each type of incomplete infection, we first provide the output of a full model, frequency = segment * modality where modality corresponds to the incomplete vs complete infection treatments. After the full tests we provide the output of per segment comparisons across modalities to identify segments whose relative frequency statistically significantly differed between incomplete and complete infections. These analyses were performed through Scheirer Ray Hare tests and Dunn tests using RStudio (packages "rcompanion" and "FSA"). The p-values indicating statistically significant differences after Bonferroni correction (p≤0.05) are in red.
(DOCX)

**S7 Table. Sequences of the primers used to quantify segment accumulation by qPCR.**
(DOCX)

**S8 Table. Statistical analysis of the comparison of DNA-R accumulation depending on the segment with which it is infiltrated.** Statistical analyses were performed through Kruskal-Wallis tests using RStudio (package "agricolae"). The p-value indicating a statistically significant difference after Bonferroni correction (p≤0.05) is in red.
(DOCX)

**S9 Table. Statistical analysis of the comparison of segment accumulation in infiltrations in pairs of segments across experimental replicates.** We provide the output of a full model, ratio = replicate * segment. Statistical analyses were performed through Scheirer Ray Hare tests using RStudio (package "rcompanion"). The p-values indicating statistically significant differences after Bonferroni correction (p≤0.05) are in red.
(DOCX)

## Acknowledgments

The authors thank Ms Sophie Le Blaye for technical help.

## Author Contributions

**Conceptualization:** Yannis Michalakis, Stéphane Blanc.

**Data curation:** Mélia Bonnamy, Andy Brousse, Yannis Michalakis.

**Formal analysis:** Mélia Bonnamy, Yannis Michalakis, Stéphane Blanc.

**Funding acquisition:** Yannis Michalakis, Stéphane Blanc.

**Investigation:** Mélia Bonnamy, Andy Brousse, Elodie Pirolles, Yannis Michalakis, Stéphane Blanc.

**Methodology:** Mélia Bonnamy, Andy Brousse, Elodie Pirolles, Yannis Michalakis, Stéphane Blanc.

**Project administration:** Yannis Michalakis, Stéphane Blanc.

**Supervision:** Yannis Michalakis, Stéphane Blanc.

**Validation:** Mélia Bonnamy, Yannis Michalakis, Stéphane Blanc.

**Visualization:** Mélia Bonnamy, Yannis Michalakis, Stéphane Blanc.

**Writing – original draft:** Mélia Bonnamy.

**Writing – review & editing:** Yannis Michalakis, Stéphane Blanc.

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
