## [Decision Letter · Decision Letter 0]

1 Nov 2023

Dear Dr. Blanc,

Thank you very much for submitting your manuscript "The genome formula of a multipartite virus is regulated both at the individual segment and the segment group levels" for consideration at PLOS Pathogens. As with all papers reviewed by the journal, your manuscript was reviewed by members of the editorial board and by several independent reviewers. The reviewers appreciated the attention to an important topic. Based on the reviews, we are likely to accept this manuscript for publication, providing that you modify the manuscript according to the review recommendations.

Regarding the extensive "Major" comment of reviewer 2: It may not be possible to address this issue experimentally, but it should at least be given careful consideration in your analysis and discussion.

Sincerely,

David M Bisaro, PhD

Academic Editor

PLOS Pathogens

Shou-Wei Ding

Section Editor

PLOS Pathogens

Kasturi Haldar

Editor-in-Chief

PLOS Pathogens

orcid.org/0000-0001-5065-158X

Michael Malim

Editor-in-Chief

PLOS Pathogens

orcid.org/0000-0002-7699-2064

Reviewer Comments (if any, and for reference):

Reviewer's Responses to Questions

**Part I - Summary**

Reviewer #1: The MS titled “The genome formula of a multipartite virus is regulated both at the individual segment and the segment group levels” builds on the previous work on the dynamics of the segmented virus faba bean necrotic stunt virus (FBNSV). The previously determined “genome formula’ of the individual components as a collective within an infection is queried here in a different manner i/e/ each segment individually with its essential partner (core module) the component that encodes the Rep protein which initiates rolling circle replication. Beyond this the authors also look at the dynamics without a subset of the segments. The authors conclude "...FBNSV genome formula is shaped by a complex process acting at both the individual segment and the segment group levels."

Certainly, this area of research has drawn attention from various non virology groups including ecologists, mathematicians, and psychologists (interested in co-operation and conflict), and will continue to do so. The MS, which is well written with good experimental design (we have to take into account that agro-infiltrations has its own challenges) has sound and modest conclusions. I do not see any issues with the MS and feel it will be well received by the scientific community at large who are thirsty for more information on the mysterious lifestyle of multicomponent viruses.

Reviewer #2: Bonnamy and coworkers present a well-written manuscript describing some exciting new results on the “genome formula” (GF) dynamics for a nanovirus, a plant DNA virus with a highly segmented genome. Infiltration assays were developed to study accumulation levels of individual genome segments, when paired with the one segment (R) that can support cellular replication of all other segments. These local-infection assays showed different accumulation levels, suggesting a bilateral process between R and other segments contributes to the unbalanced GF. However, since these bilateral comparisons did not predict the overall GF in systemic infection, the authors also considered the GF in systemic infections missing non-essential segments. As these GF values varied in some cases, the authors conclude that complex interactions between multiple segments during systemic infection likely shape the GF.

This paper follows up previous work in this model system, but introduces new approaches, results and insight. Overall, I think the presentation of the results is clear and balanced, and the results are highly interesting and worthy of publication. However, I do have a methodological concern and some minor comments on the analysis and presentation. I hope the authors find these comments useful for improving this work.

**Part II – Major Issues: Key Experiments Required for Acceptance**

Reviewer #1: none

Reviewer #2: (No Response)

**Part III – Minor Issues: Editorial and Data Presentation Modifications**

Reviewer #1: I am wondering if it may be worth mentioning that Agrobacterium tumefaciens is now known as Agrobacterium radiobacter. Line 364 – optical density should be all small letters.

Reviewer #2: Major comment:

If I have understood the setup correctly, the authors infiltrate leaves with multiple transgenic agrobacterium strains, each of which introduces a genome segment. The authors also indicate at various points throughout the manuscript that the infiltration efficiency is variable, which is normal for this procedure. The authors therefore normalize the accumulation of the segment in question (call it X) with the level of accumulation of R. However, given that the two segments are being introduced by separate bacterial strains, this means the cells infiltrated by the two strains have different spatial distributions. Again, this point is acknowledged clearly by the authors, for example in the conclusion (e.g. lines 303-306, here referring to infiltrations with all genome segments). So that leaves us with uninfiltrated cells, fraction P(no R, no X), cells infiltrated with one of the two segments, fractions P(no R, X) or P(R, no X), and cells infiltrated with both segments, fraction P(R, X). The fractions P(no R, no X) and P(no R, X) will not contribute to accumulation, as R is required. Both the fractions P(R, no X) and P(R, X) will contribute to the measured accumulation, but the fraction P(R, no X) obviously only produces R and never X. The authors normalize the level of X by R, but that disregards the fact that only the fraction p(R, X) can contribute to the accumulation of X. Assuming random distributions, the fraction p(R, X) will be the product of subpopulations p(R)p(X). Therefore, changes in infiltration efficiency can have a much stronger effect on the fraction of p(R, X) than accounted for by a correction with p(R).

As a concrete example: imagine the accumulation of X is always equal to that of R in co-infiltrated cells. When we have a low level of infiltration efficiency (say < 50% of all susceptible cells for each bacterial strain), the fraction of p(R, X) is much smaller than the fraction p(R, no X), so by normalizing by R we grossly underestimate the actual accumulation of X in co-infiltrated cells (0.25 of the accumulation of R). When infiltration efficiency is very high (99%), after normalizing we get the correct value (~ 1).

In an ideal world, pairs of segments would be introduced by a single bacterial strain (introduces many complications) or an experimental estimate of the fraction p(R, X) would be obtained (also quite complex, e.g. requiring a separate treatment e.g. with fluorescent markers, leading to indirect comparisons). As neither of these options are very feasible, some remedies I can think of are: (i) show that infiltration efficiency is always relatively high and therefore its impact is limited, (ii) consider how much variation there is in R accumulation, and in particular whether this variation is segment dependent (where if it is segment dependent, it does complicate the interpretation of the results as there could be systematic biases which are amplified due to the reliance on the product p(R)p(X) in the observed accumulation), or (iii) provide some evidence the R protein is present in most of the infiltrated cells because of the viral gene expression product sharing (i.e., Sicard et al. 2019) (from reading the authors comments in the discussion, they do not believe this is the case). To be clear, to my mind this point does not invalidate the research, but it should be given careful consideration in the analysis and interpretation of the results, where possible.

Minor comments:

1. I find the Author Summary (AS) a clear description of the research, as the abstract remains a little too abstract for me. Perhaps the authors can use the AS as a basis for the abstract?

2. Lines 93-95: “The eight FBNSV segments are always found [...]”. That statement is undoubtedly correct, but what is the sample size it is based on? And consequently, how much weight can we lend to it?

3. Related to the major comment and a discussion point: What about the effects of other segments on the accumulation of R? I would not be surprised if some viral protein-protein interactions do lead to e.g. a higher productivity of the M-Rep? Viral PPI are mentioned later in the discussion, but not in the section discussing differences in segment accumulation (lines 245-259).

4. The GF reported in Figure 1 for systemic leaves correlates quite nicely with previous reports from this group. It may be worthwhile pointing this out, as I for one immediately want to cross-reference them.

5. Lines 282-283, point II: “the ABSENCE of interactions between the different segments beyond that with R” would account for the differences between the bilateral vs. communal comparisons? I would think its the opposite way around and that it is precisely these interactions that could account for the differences.

6. Just by looking at the GF data, there appears to be quite some heterogeneity in variances. However, ANOVA and Tukey posthoc tests are reported throughout the analysis. I see that these procedures are performed on transformed data, but I really wonder whether homoscedasticity assumptions are being met. Good to show this is the case and, if not, to adjust the procedure accordingly.

PLOS authors have the option to publish the peer review history of their article (what does this mean?). If published, this will include your full peer review and any attached files.

Reviewer #1: No

Reviewer #2: **Yes: **Mark P. Zwart

Figure Files:

Data Requirements:

Reproducibility:

References:

---

## [Editor Report · Decision Letter 1]

14 Jan 2024

Dear Dr. Blanc,

We are pleased to inform you that your manuscript 'The genome formula of a multipartite virus is regulated both at the individual segment and the segment group levels' has been provisionally accepted for publication in PLOS Pathogens.

Best regards,

David M Bisaro, PhD

Academic Editor

PLOS Pathogens

Shou-Wei Ding

Section Editor

PLOS Pathogens

Kasturi Haldar

Editor-in-Chief

PLOS Pathogens

orcid.org/0000-0001-5065-158X

Michael Malim

Editor-in-Chief

PLOS Pathogens

orcid.org/0000-0002-7699-2064
---

## [Editor Report · Acceptance letter]

22 Jan 2024

Dear Dr. Blanc,

We are delighted to inform you that your manuscript, "The genome formula of a multipartite virus is regulated both at the individual segment and the segment group levels," has been formally accepted for publication in PLOS Pathogens.

Best regards,

Michael Malim

Editor-in-Chief

PLOS Pathogens

orcid.org/0000-0002-7699-2064